# Pickering Emulsion Stabilized by β-Cyclodextrin and Cinnamaldehyde/β-Cyclodextrin Composite

**DOI:** 10.3390/foods12122366

**Published:** 2023-06-14

**Authors:** Caihua Liu, Yachao Tian, Zihan Ma, Linyi Zhou

**Affiliations:** 1College of Food and Health, Beijing Technology and Business University, Beijing 100048, China; 17686963949@163.com (C.L.); tianyachaopaper@163.com (Y.T.); 2College of Food Science, Northeast Agricultural University, Harbin 150030, China; mzh6777@163.com

**Keywords:** food emulsion, storage stability, bioactive substances, antioxidation, digestion

## Abstract

A Pickering emulsion was prepared using β-cyclodextrin (β-CD) and a cinnamaldehyde (CA)/β-CD composite as emulsifiers and corn oil, camellia oil, lard oil, and fish oil as oil phases. It was confirmed that Pickering emulsions prepared with β-CD and CA/β-CD had good storage stability. The rheological experiments showed that all emulsions had G′ values higher than G″, thus confirming their gel properties. The results of temperature scanning rheology experiments revealed that the Pickering emulsion prepared with β-CD and CA/β-CD composites had high stability, in the range of 20–65 °C. The chewing properties of Pickering emulsions prepared by β-CD and corn oil, camellia oil, lard, and herring oil were 8.02 ± 0.24 N, 7.94 ± 0.16 N, 36.41 ± 1.25 N, and 5.17 ± 0.13 N, respectively. The chewing properties of Pickering emulsions made with the CA/β-CD composite and corn oil, camellia oil, lard, and herring oil were 2.51 ± 0.05 N, 2.56 ± 0.05 N, 22.67 ± 1.70 N, 3.83 ± 0.29 N, respectively. The texture properties confirmed that the CA/β-CD-composite-stabilized-emulsion had superior palatability. After 28 days at 50 °C, malondialdehyde (MDA) was detected in the emulsion. Compared with the β-CD and CA + β-CD emulsion, the CA/β-CD composite emulsion had the lowest content of MDA (182.23 ± 8.93 nmol/kg). The in vitro digestion results showed that the free fatty acid (FFA) release rates of the CA/β-CD composite emulsion (87.49 ± 3.40%) were higher than those of the β-CD emulsion (74.32 ± 2.11%). This strategy provides ideas for expanding the application range of emulsifier particles and developing food-grade Pickering emulsions with antioxidant capacity.

## 1. Introduction

Traditional emulsions are dispersions of two immiscible liquids that are thermodynamically unstable and need to be stabilized by surfactants [1,2]. As a vital part of the industry, emulsions have been widely used in food [3,4,5], medicine [1,6,7], cosmetics [8,9,10], and other industries. As an unstable system, emulsions spontaneously separate during storage [1,2,11,12]. Emulsion stability can be improved by adding a stabilizer [13]. Traditional emulsions improve their stability by adding many surfactants [14]. Surfactants have low biocompatibility, environmental pollution, and toxicity [8,15]. These disadvantages severely limit surfactant application in emulsions [2]. To improve emulsion stability and safety, it is urgent to find safer and more reliable emulsifiers.

In contrast to traditional emulsions, Pickering emulsions are stabilized by solid particles, making them well suited to food applications [2]. The particles that can be used as Pickering emulsion stabilizers come from a wide range of sources, including inorganic particles, synthetic particles, and organic particles. Inorganic particles are the earliest stabilizers used in Pickering emulsions, including clay [16], calcium carbonate [17], silica [7,18,19], etc. Synthetic particles include various synthesized polymers [20,21]. Inorganic and synthetic particles are less used in the food fields due to their poor safety and biocompatibility [22]. Thus, natural organic particles have become the best choice for preparing food-grade Pickering emulsions [15]. Natural organic particles are often used in Pickering emulsions. These include polysaccharides [8], proteins [12], lipid particles [9], and their derivatives. Previous studies reported food-grade Pickering emulsions prepared with polysaccharide particles (such as starch [23,24], chitosan [25], cellulose [26,27], etc.), lipid-based particles (such as phospholipids [28], monoglycerides [29], etc.), and protein particles (such as zein [30,31,32,33], soy protein isolate [34,35,36], ovalbumin [37,38], etc.). Earlier studies indicated that the relative molecular weight of particles was inversely proportional to their emulsifying ability [23]. Although the safety of Pickering emulsions prepared from these natural particles meets the requirements of food applications, their molecular masses are so large that Pickering emulsions have poor storage stability [2,39]. To obtain a food-grade Pickering emulsion with high safety and stability, small-molecule natural products and their derivative particles as stabilizers must be sought.

β-cyclodextrin (β-CD) is a small molecular natural starch derivative that is produced by the action of cyclodextrin glucosyltransferase on starch [40,41]. It consists of seven glucoses linked by α-1,4-glycosidic bonds, forming a cyclic oligosaccharide [2,42]. The outer surface of β-CD is hydrophilic, and the inner cavity is hydrophobic, which facilitates the encapsulation of hydrophobic substances [2]. Generally, substances embedded in β-CD are not only more stable when exposed to light, heat, and oxygen but also improve color, fragrance, and flavor [43,44,45]. Therefore, it has wide applications in the food [46], cosmetic [47], and pharmaceutical [48,49] industries. Its amphiphilicity makes it useful for stabilizing Pickering emulsions by adsorbing at the oil–water interface [50]. Its emulsifying ability is much better than protein, starch, and lipid particles. Shimada [51] and Durante [52] investigated the stability of an emulsion formed with α-cyclodextrin, β-CD, and γ-cyclodextrin and found that β-CD had the highest emulsifying effect. Due to its excellent safety and biocompatibility, β-CD as an effective emulsifier has been the focus of intense food field research [53,54,55]. Compared with other types of food, lipids in emulsion food are more easily oxidized. This is because there are oxides such as dissolved oxygen and metal ions in the continuous phase. Kibici and Kahveci found that the addition of β-CD can improve the storage stability of a Pickering emulsion, but it cannot improve the oxidation stability of the emulsion [56]. Lipid oxidation can lead to unpleasant flavors, color changes, loss of nutritional value, and even potentially toxic substances. Therefore, it is a challenge to develop an antioxidant-loaded β-CD to meet the excellent oxidative stability of emulsions.

Cinnamaldehyde (CA) is a hydrophobic active matter isolated from natural Cinnamomum cassia that is volatile and has a unique fragrance [57]. CA has bactericidal, antiviral, and antioxidative effects [42,58]. To improve Pickering emulsion antioxidant properties, fat-soluble antioxidants are usually added directly to the oil phase [59]. Due to lipid oxidation in Pickering emulsions occurring at the oil-water interface, adding antioxidants directly to the oil phase cannot achieve effective antioxidant properties. It is essential to introduce CA to the oil–water interface film in order to solve the antioxidant problem of Pickering emulsions. As a result of the stabilization of Pickering emulsions with CD and vitamin E, the stability and bioaccessibility of vitamin E have been enhanced, as well as the antioxidant properties of the Pickering emulsions [39]. In order to create an oil–water interface with antioxidant function, a CA/β-CD composite as an emulsifier was used to disperse the oil at a high speed. This strategy brings antioxidant-active substances to the oil–water interface, giving the emulsion excellent and long-lasting antioxidant properties. Additionally, the long-term storage stability, thermodynamic stability, texture properties, and digestibility of emulsions are important factors to consider.

The interfacial mechanism of β-CD- and CA/β-CD-stabilized Pickering emulsions was investigated in our previous studies. However, their application in different oil phases and the antioxidant and simulated digestibility of their prepared emulsions still need to be studied. In this study, a Pickering emulsion was prepared using β-CD and a CA/β-CD composite. The emulsifying ability of β-CD and the CA/β-CD composite for corn oil, camellia oil, lard, and herring oil was evaluated by considering the storage stability and thermodynamic stability of the Pickering emulsion. At the same time, the digestibility and antioxidant capacity of the Pickering emulsion were evaluated by a simulated digestion experiment and an antioxidant experiment. The research results can be applied to the development of new emulsion food and health products.

## 2. Materials and Methods

### 2.1. Materials

Corn oil (>98%) was obtained from Kele Fine Chemical Co., Ltd. (Wuhan, China). Camellia oil (>99%) was obtained from Baicao Pharmaceutical Co., Ltd. (Jian, China). Herring oil (>99%) was obtained from Yongkuo Technology Co., Ltd. (Tianmen, China). Lard was obtained from Angel Yeast Co., Ltd. (Yichang, China). Silicone oil purchased from Boshi Chemical Co., Ltd. (Wuhan, China). Cinnamaldehyde (>97%) and β-CD (>99%) purchased from Macklin Biochemical Technology Co., Ltd. (Shanghai, China). An MDA test kit was purchased from Solarbio Science & Technology Co., Ltd. (Beijing, China). N-hexane was purchased from Damao Chemical Reagent Co., Ltd. (Tianjin, China). All other reagents were analytical grade.

### 2.2. Preparation and Characterization of CA/β-CD Composite

X-ray diffractometer (XRD) and Fourier-transform infrared spectrometer (FTIR) were both used to confirm the successful preparation of inclusion complexes in our previous studies [2].

### 2.3. Preparation of the Pickering Emulsion

#### 2.3.1. Preparation of the Pickering Emulsion

Pickering emulsion was prepared by using corn oil, camellia oil, lard, and herring oil as the oil phase, and β-CD and CA/β-CD inclusion compounds were used as an emulsifier. Our previous experimental conditions were slightly modified to determine the Pickering emulsion’s preparation parameters [2]. In short, the addition of β-CD and CA/β-CD composites are 3% (*w*/*v*), the oil phase is 60% (*v*/*v*), the dispersion time is three minutes, and the disperser speed is 25,000 rpm. The homogenizer type is LC-SFJ-10, Lichen Technology Co., Ltd. (Shanghai, China). At room temperature, the Pickering emulsions were stored (28 days) for further analysis.

#### 2.3.2. Storage Stability, Particle Size, and Zeta Potential of Pickering Emulsion

At room temperature, the emulsion samples were stored for four weeks in sample bottles. The creaming index (*CI*) was used to determine the storage stability of the emulsions, and it was calculated by the following formula:(1)$$CI\%=\left(He/Ht\right)\times 100$$
where *He* and *Ht* represent the height of the serum and the total height of the samples.

Freshly prepared emulsion samples were measured using dynamic light scattering (DLS, Malvern 3000 Instruments, Worcestershire, UK). After 150-times dilution, the samples were measured. At 25 °C, the oil’s refractive index and absorption index were 1.45 and 0.001, while the water’s were 1.33 and 0. The results were reported as the averages of three or more readings.

#### 2.3.3. Measurements of Texture Properties

The hardness, chewing properties, and cohesion of the prepared Pickering emulsion samples were analyzed by a Brookfield CT3 texture analyzer (Brookfield, WI, USA). Select the TPA mode for detection, and the specific parameters are set as follows: falling speed of 2 mm/s, detection speed of 0.5 mm/s, rising speed of 2 mm/s, trigger force of 2.0 g, and compression distance of 1 cm.

#### 2.3.4. Measurements of Rheological Properties

A rheometer (Anton Paar, Graz, Austria) was used to measure the rheological properties of emulsion samples. During the experiment, a gap of 1 mm was used with the parallel plate probe PP50. Firstly, the linear viscoelastic range of the sample was determined via a strain experiment (frequency, 1 Hz; strain range, 0.01–100%). In the linear viscoelastic region, the rheological properties of Pickering emulsions were tested with a temperature-scanning module. The temperature scanning ranged between 20 °C and 90 °C, with a temperature change rate of 5 °C/min. In order to prevent water evaporation, the edge of the parallel plate was sealed with silicone oil.

#### 2.3.5. Fluorescence Microscope Observation

The Pickering emulsions were prepared using corn oil as the oil phase, β-CD and CA/β-CD as the emulsifiers. The addition of β-CD and the CA/β-CD composite is 3% (*w*/*v*), the oil phase is 60% (*v*/*v*), the dispersion time is 3 min, and the dispersion speed is 25,000 rpm. Labeled emulsions were observed under a fluorescent microscope (Olympus, Tokyo, Japan). The Pickering emulsion staining method is as follows: 300 μL of Nile blue (0.1%, *w*/*v*) and Nile red (0.01%, *w*/*v*) dye solution were added to 10 mL of sample and stained in the dark for 30 min.

#### 2.3.6. Measurement of Oxidation Resistance

The Pickering emulsions were prepared using corn oil as the oil phase and β-CD and CA/β-CD as the emulsifiers. The addition of β-CD and the CA/β-CD composite is 3% (*w*/*v*), the oil phase is 60% (*v*/*v*), the dispersion time is 3 min, and the dispersion speed is 25,000 rpm. The emulsion samples were stored at 50 °C for 28 days. The antioxidant activity of the emulsion was evaluated by the content of secondary oxide malondialdehyde (MDA) in the emulsion. The content of MDA was determined by the thiobarbituric acid method. By improving the method of Cheng-Ye Zhan et al. [60], the detection method for MDA content was determined. The principle is that MDA and thiobarbituric acid can generate a red-brown substance called trimethoprim under acidic and high-temperature conditions. MDA determination is interfered with by many substances, the most important of which is soluble sugar. The maximum absorption wavelength of the color reaction product of sugar and TBA is 450 nm, but 532 nm is also absorption. Therefore, measure the absorbance at 600 nm, 532 nm, and 450 nm at the same time. Use the difference between the absorbance at 532 nm and 450 nm and 600 nm to calculate the MDA concentration. The absorbance values at 600 nm, 532 nm, and 450 nm were measured with a microplate reader to calculate the MDA content. In order to obtain accurate data, the experiment was carried out using the MDA detection kit. MDA content is calculated as follows:(2)$$MDA\left(nmol/g\right)=\frac{6.45\times \left(A532-A600\right)-0.56\times A450}{M}$$
where *A*450, *A*532, and *A*600 are the absorbance values at 450 nm, 532 nm, and 600 nm, respectively. Coefficients of 6.45 and 0.56 are used to exclude the influence of soluble sugars. *M* is the weight of the emulsion.

#### 2.3.7. In Vitro Simulated Digestion of Emulsion

The Pickering emulsions were prepared using corn oil as the oil phase and β-CD and CA/β-CD as the emulsifiers. The addition of β-CD and the CA/β-CD composite is 3% (*w*/*v*), the oil phase is 60% (*v*/*v*), the dispersion time is 3 min, and the dispersion speed is 25,000 rpm. With appropriate modifications, simulated digestive juices were prepared according to the method of Camila Mella et al. [61]. Simulated gastric fluid (SGF): 0.02 g NaCl and 0.32 g porcine pepsin were dissolved in 100 mL phosphate buffer (10 mmol/L), and the pH was adjusted to 2.0 with HCl solution (0.1 mol/L). The simulated gastric digestion experiment was carried out at 37 °C by adding 10 mL of simulated gastric juice to 5 mL of fresh emulsion. The droplet morphology changes of the emulsions during digestion were observed via an optical microscope.

Simulated intestinal solution (SIF): potassium dihydrogen phosphate 6.8 g, porcine pancreatic lipase 10 g, and pig bile salt 5 g, dissolved in an appropriate amount of water. The pH value of the SIF was adjusted to 6.8 with NaOH (0.2 mol/L) or HCl (0.2 mol/L). After simulated gastric digestion, 15 mL of simulated intestinal solution was directly added. The temperature was maintained at 37 °C. NaOH (0.1 mol/L) solution was used to maintain the pH value of the mixture at 6.8. An optical microscope was used to observe the morphological changes of the samples during simulated digestion. The release rate of free fatty acids during digestion was determined according to the consumption of NaOH. The formula is as follows:(3)$$FFA\left(\%\right)=\frac{V\times C\times M}{2m}\times 100$$
where *C* represents the concentration of NaOH (0.1 mol/L), *V* (mL) represents the volume of NaOH consumed, *M* (g/mol) represents the molar mass of oil, 2 represents one molecule of triglyceride producing two molecules of FFA, and *m* (g) represents the mass of oil in the added emulsion.

#### 2.3.8. Statistical Analysis

Samples were tested three times. Data were summarized as mean values ± standard deviations. An analysis of variance (ANOVA) and Tukey test were used to analyze all experimental data, using SPSS (Chicago, IL, USA), and a *p* < 0.05 was considered to be statistically significant.

## 3. Results and Discussion

### 3.1. Analyses of Storage Stability and Particle Size and Zeta Potential

The storage stability of Pickering emulsion is a critical index to measure the emulsifying ability of emulsifier particles. There are significant differences in the types and quantities of fatty acids contained in different types of oils [62]. These differences may have a substantial impact on the performance of Pickering emulsions. Figure 1 shows the storage stability of Pickering emulsions formed with β-CD (I) and a CA/β-CD composite (II) with corn oil (a), camellia oil (b), lard (c), and herring oil (d) after a storage period of 28 days at room temperature. It can be seen from Figure 1 that β-CD and its complexes have the ability to emulsify different oil phases to form Pickering emulsions with long-term storage stability. The *CI* of all emulsions is 0%. The Pickering emulsion prepared by emulsifying different oil phases with β-CD and its inclusion compound as emulsifiers did not break during the 28-day storage. The storage stability of a food-grade Pickering emulsion is closely related to its shelf life. Emulsions with better storage stability have a longer shelf life. Emulsions with a longer shelf life are more desirable, as they have a greater chance of being sold and used before the expiration date. This shows that β-CD and its complexes can be widely used in the development of various food-grade emulsions, and this is of substantial significance for expanding their application range [63,64,65,66].

The droplet size and ζ-potential of the Pickering emulsions formed with β-CD and the CA/β-CD composite with corn oil, camellia oil, lard, and herring oil are show in Table 1. In the emulsion prepared with β-CD, the droplet sizes of the corn oil, camellia oil, and herring oil emulsions were 10.52 ± 0.75 μm, 11.92 ± 0.63 μm, and 10.86 ± 1.15 μm, respectively. In the emulsion prepared with CA/β-CD, the droplet sizes of the corn oil, camellia oil, and herring oil emulsions were 11.02 ± 0.45 μm, 11.67 ± 0.40 μm, and 10.67 ± 0.87 μm, respectively. The β-CD- and CA/β-CD-stabilized Pickering emulsions prepared from the same oil phase had similar average particle sizes. It was confirmed in our previous study that the added amount of β-CD and CA/β-CD is the main factor affecting emulsion particle size [2]. β-CD and its compound were added in the same amount, and the emulsion particles were also similar in size. The lard emulsion had the largest average particle size among the emulsions stabilized by β-CD and the CA/β-CD composite. The particle sizes were 25.57 ± 5.87 μm and 27.39 ± 6.67 μm, respectively. Different oils contain different amounts of saturated fatty acids. Saturated fatty acids make up 42% of lard oil, 20% of herring oil, 14% of corn oil, and 6% of camellia oil. The higher the saturated fatty acid content, the easier it is to solidify the oil. In terms of the melting temperature, lard oil is around 32–49 °C [67], herring oil is about 10 °C [68], corn oil is −11 °C [69], and camellia oil is −17 °C [70]. At room temperature, the lard emulsion coagulates into a solid, resulting in larger droplets. The results of the ζ-potential values showed that all samples were negatively charged. The higher the absolute value of the ζ-potential, the better the Pickering emulsion’s stability [71]. In the emulsion prepared with β-CD, the ζ-potentials of the corn oil, camellia oil, lard and herring oil emulsions were −32.31 ± 2.47 mV, −30.56 ± 0.89 mV, −23.20 ± 1.32 mV, and −1.33 ± 1.66 mV, respectively. In the emulsion prepared with CA/β-CD, the ζ-potentials of corn oil, camellia oil, lard and herring oil emulsion were −32.36 ± 1.57 mV, −33.40 ± 1.04 mV, −21.37 ± 1.71 mV, and −29.67 ± 1.37 mV, respectively. The ζ-potential results showed that the lard emulsion’s stability might be lower than that of other emulsions. In contrast to that of the lard emulsion, all other emulsion potentials are close to 30, indicating excellent storage stability [2].

### 3.2. Analyses of Texture Properties

Texture properties are a critical factor in evaluating emulsions. They directly affect the oral behavior, flavor release, and sensory pleasure of the emulsion. This is a major factor affecting consumers’ preference for and acceptability of emulsions. Figure 2 shows the results of testing the hardness of emulsion samples at room temperature. The hardness of Pickering emulsions prepared by β-CD and corn oil, camellia oil, lard, and herring oil were 18.12 ± 0.85 N, 17.02 ± 0.62 N, 81.67 ± 3.86 N, and 10.57 ± 1.87 N, respectively. The hardness of Pickering emulsions prepared with CA/β-CD composite and corn oil, camellia oil, lard, and herring oil were 6.03 ± 0.05 N, 6.33 ± 0.85 N, 49.23 ± 4.84 N, and 8.13 ± 0.62 N, respectively. Under the same oil-phase conditions, the β-CD emulsion has a higher hardness than the CA/β-CD composite emulsion. This difference is caused by the thicker oil–water interface film of the β-CD emulsion [2]. The CA/β-CD inclusion complex contains the hydrophobic substance CA, which will increase its intermolecular repulsion [2]. This will hinder the crystallization and aggregation of the inclusion complex at the emulsion-droplet interface film [2], and it will reduce the hardness of the emulsion and increase palatability. Under the same emulsifier conditions, the lard emulsion has the highest hardness compared to other oil emulsions. This is because lard has a high freezing point. The lard emulsion is solid at room temperature, while other emulsions are gels.

Figure 3 shows the chewing-property diagram of different types of Pickering emulsion. Pickering emulsions prepared with corn oil, camellia oil, lard, and herring oil showed chewing properties of 8.02 ± 0.24 N, 7.94 ± 0.16 N, 36.41 ± 1.25 N, and 5.17 ± 0.13 N, respectively. The chewing property of Pickering emulsions prepared with the CA/β-CD composite and corn oil, camellia oil, lard, and herring oil were 2.51 ± 0.05 N, 2.56 ± 0.05 N, 22.67 ± 1.70 N, and 3.83 ± 0.29 N, respectively. Similar to Figure 2, under the same oil-phase conditions, the β-CD emulsion has a higher chewing property than the CA/β-CD composite emulsion, and the lard emulsion has the highest chewing property under different oil-phase conditions. There is a positive correlation between hardness and chewiness. The higher the hardness of the food, the more effort it takes to chew [72].

Cohesion can reflect the ability of the emulsion to restore its original structure after being subjected to external mechanical pressure [73]. Strong cohesion increases the emulsions’ recovery ability. Figure 4 is the cohesion diagram of different types of Pickering emulsions. Compared with the results of the hardness and chewiness experiments, the cohesiveness of different types of emulsions showed opposite results. The cohesion of the Pickering emulsions prepared by β-CD and corn oil, camellia oil, lard, and herring oil were 0.75 ± 0.04, 0.77 ± 0.02, 0.42 ± 0.01, and 0.79 ± 0.07, respectively. Moreover, the cohesion of the Pickering emulsions prepared by CA/β-CD composite and corn oil, camellia oil, lard, and herring oil were 0.86 ± 0.07, 0.84 ± 0.01, 0.59 ± 0.01 N, and 0.83 ± 0.01, respectively. Under the same oil-phase conditions, the CA/β-CD composite emulsion has a higher cohesion than the β-CD emulsion, and the lard emulsion has the lowest cohesion under different oil-phase conditions. The reason is that the introduction of the hydrophobic molecule CA into the cavity of β-CD results in an increase in the repulsion between β-CD molecules [2,42]. This prevents β-CD molecules from approaching each other and crystallizing [2]. Excessive composites were distributed in the emulsion to form a similar network structure, which improved the self-recovery ability of the emulsion [2]. These results suggest that the Pickering emulsion prepared with the CA/β-CD composite is softer and has better palatability, making it suitable to be applied with β-cyclodextrin in different emulsion food systems.

### 3.3. Analyses of Rheological Properties

The rheological properties of the emulsion under the temperature-scanning module can reflect the microstructure of the emulsion. In traditional emulsions, the continuous phase often determined the rheological properties [2]. In Pickering emulsions, oil phases would have a greater impact because oil droplets are closer together [74]. Figure 5 shows that all samples have higher G′ values than G″ values, confirming their gel-like elastic properties [75]. When the temperature increased from 20 °C to 90 °C, the G′ and G″ values of Pickering emulsions prepared by emulsifier particles (β-CD and CA/β-CD composite) and lard decreased rapidly with the increase in temperature. Due to the higher freezing point of lard, the initial temperature of the temperature-scanning experiment was 20 °C, and the emulsion was solid [76]. As the emulsion changes from liquid to solid, the droplet volume increases, and the protective film at the oil–water interface breaks. As the temperature increased, the lard emulsion changed from a solid to a liquid state, decreasing the G′ and G″ values [77]. In addition, as the temperature rose, the emulsifier particles that were adsorbed at the oil–water interface began to fall off, destroying the emulsion’s basic structure and rapidly reducing its G′ and G″ values [78].

The changes in the G′ and G″ values of Pickering emulsions prepared by the other three oil phases were significantly different from those of the lard emulsion. As the temperature rose from 20 °C to 65 °C, the G′ value and G″ value of the corn oil emulsion, camellia oil emulsion, and herring oil emulsion increased. Moderate heat treatment can partially dissolve the emulsifier particles adsorbed at the surface of the emulsion droplets [79]. A network structure formed by excessive emulsifier particles will also partially dissolve [2]. The Pickering emulsion’s original ordered structure was disrupted, and the surface of the emulsion droplets became rougher [79]. The emulsion flow is prevented by increased intermolecular friction. In addition, limited heat treatment does not destroy the basic structure of the emulsion [80], so as temperatures rise from 20 °C to 65 °C, the G′ value and G″ value gradually increase. The results showed that the emulsions prepared with the β-CD and CA/β-CD complex had an excellent thermal stability. As the temperature rises from 65 °C to 90 °C, the G′ value and G″ value of the emulsion decrease. Due to too-high temperatures, emulsifier particles will move violently and gradually fall from the oil–water interface, and the original complete network structure will also degrade [81].

In the temperature-drop stage, the G′ and G″ values of lard emulsion change primarily due to the change in the lard state from liquid to solid. In Figure 5, A2 and B2 indicate that the elastic modulus and loss modulus of the lard emulsion increased slowly as the temperature decreased from 90 °C to 25 °C. When the temperature decreased from 25 °C to 20 °C, the elastic modulus and loss modulus of the lard emulsion increased exponentially. It can be determined that between 25 and 20 °C, the lard emulsion changed from a liquid to a solid. After preparing the lard for the lard emulsion, the freezing point of the lard decreased significantly, which is of critical significance for the development of new lard foods.

As the temperature decreased from 90 °C to 65 °C, the G′ value and G″ value of β-CD emulsion increased more than that of the CA/β-CD composite emulsion. This is due to the fact that molecular motion is active in this temperature range. As the temperature decreases, the emulsifier particles dissolved in the aqueous phase are precipitated [82]. The β-CD molecules will spontaneously adsorb and aggregate at the oil–water interface, which can repair the interface film. However, due to the introduction of hydrophobic substances in the CA/β-CD composite, the intermolecular repulsion increases, thus preventing the adsorption of CA/β-CD composite particles at the oil–water interface. From 65 °C to 20 °C, the rough surface of the emulsion droplets became smooth again, the disordered structure became orderly, and the G′ and G″ values of the emulsion decreased gradually [83]. Efficient thermal stability is a key factor in the sales and promotion of emulsion food [84]. The results of temperature scanning rheology experiments revealed that the Pickering emulsion prepared with β-CD and CA/β-CD composites had high stability in the range of 20–65 °C.

### 3.4. Analyses of Microstructure

The microstructure of emulsion droplets can be observed using fluorescence-staining microscopes. Figure 6 shows fluorescent-staining images of Pickering emulsions prepared with corn oil as the oil phase and β-CD and CA/β-CD as emulsifiers. Oil is shown in green, and emulsifiers are shown in red. The figure shows that excess β-CD particles crystallize and aggregate on the surface of emulsion droplets, forming a thicker interface protective film. A similar network structure is formed by excess CA/β-CD complex particles in the aqueous phase. The fluorescence-staining images directly reveal the phenomenon that β-CD molecules aggregate on the surface of oil droplets to form a thick protective film, causing the emulsion droplets to become larger. There was no obvious crystallization in the CA/β-CD composite emulsion droplets. It can be clearly seen that excessive CA/β-CD molecules are not aggregated at the oil–water interface but dispersed in the continuous phase. On the one hand, these inclusion complex molecules dispersed in the continuous phase prevent the aggregation between droplets and increase the stability of the emulsion [2]. On the other hand, they play a role in lubrication and improve the texture of the emulsion [2]. These findings provide evidence for explaining the differences in texture and rheological properties between the β-CD and CA/β-CD composite emulsions.

### 3.5. Analyses of Antioxidant Capacity

Malondialdehyde (MDA) is the end product of lipid peroxidation caused by oxygen free radicals or metal ions. The antioxidant capacity of a Pickering emulsion can be determined by detecting the content of MDA in the emulsion [85]. For comparison, Pickering emulsions stabilized by β-CD, the CA/β-CD composite, and β-CD + CA (CA was added to the oil phase) were prepared. Corn oil was the oil phase in all emulsion samples. In the first five days of the antioxidant experiment, the MDA content in the emulsion stabilized by CA + β-CD was lower than that in the CA/β-CD composite. According to Figure 7a, the β-CD emulsion had the highest content of MDA (569.41 ± 9.37 nmol/kg) after storage for 28 days. The Pickering emulsion was stabilized by β-CD + CA, and the MDA content was reduced to 380.54 ± 8.44 nmol/kg. Meanwhile, the CA/β-CD-stabilized emulsion had the lowest content of MDA (182.23 ± 8.93 nmol/kg). Compared with β-CD + CA emulsion, the CA/β-CD composite emulsion had longer antioxidant activity and sustained release effect. The result might be attributed to the fact that the CA/β-CD emulsifier carried the CA to the oil–water interface, where the CA could effectively diminish lipid oxidation [39]. Studies have shown that the oil–water interface is the most significant place for lipid oxidation in emulsions [86]. β-CD can help to stabilize emulsions, preventing them from separating into their oil and water components [56]. However, it cannot protect emulsions against oxidation, as it does not have antioxidant properties [56]. Therefore, other antioxidants must be used to protect emulsions from oxidation. Thus, the most effective method to solve lipid oxidation in emulsions is to design oil–water interface films with antioxidant properties.

### 3.6. Analyses of In Vitro Digestion

The digestion and absorption characteristics of fat-soluble nutrients in emulsion food are very important for the promotion and application of products [39]. Figure 8 shows the digestion properties of emulsions. In all samples of emulsion, corn oil was the oil phase. The evidence can be seen in the optical microscope images that both β-CD and CA/β-CD composite emulsions can maintain high stability in simulated gastric juice. This may be because the oil–water interface protective shell of the Pickering emulsion enables the emulsion to remain stable in SGF. The digestion and absorption of the emulsion is mainly carried out in simulated intestinal fluid (SIF). Under SIF conditions, the number of emulsion drops was significantly reduced after 60 min, indicating that the lipid colloids were digested in large quantities at this stage. Figure 9 shows the release rate of FFA from the Pickering emulsions stabilized by β-CD and the CA/β-CD composite in SIF digest. In SIF, the release rate of FFA in emulsions revealed the same phenomenon. The highest release rate of FFA was 74.32 ± 2.17% in the β-CD emulsion and 87.49 ± 3.43% in CA/β-CD composite emulsion. The FFA release rate of the CA/β-CD composite emulsion was higher and faster than that of the β-CD emulsion. Due to the thicker protective shell of the CD emulsion droplets, oil droplets have less contact area with lipase.

## 4. Conclusions

In summary, a novel emulsifier particle (CA/β-CD composite) with antioxidant function was prepared by assembling CA and β-CD. β-CD and the CA/β-CD composite as emulsifiers can be applied in many different oil phases. Under the same oil phase conditions, Pickering emulsions prepared with BCD and CA/BCD have similar particle sizes. Within 28 days, all prepared emulsions were not layered. The freezing point of the oil phase has a major influence on the texture properties and thermal stability of the emulsion. The CA/β-CD composite emulsion has lower hardness and chewiness than the β-CD emulsion and higher cohesion than the β-CD emulsion due to its thinner protective shell and network structure formed in the continuous phase. Through the interfacial adsorption effect, the CA/β-CD composite emulsion can bring substances with antioxidant activity to the oil–water interface. This strategy can be used to design functional emulsions with much higher antioxidant capacities than conventional emulsions. Both the β-CD and CA/β-CD composite emulsions had good stability in simulated gastric digestion, and fatty acids were mainly released in a simulated intestinal environment. This result provides a new way to transport active substances by using a Pickering emulsion as the carrier. In addition, the study also provides an innovative path for the design of emulsifier particles and the development of functional emulsion foods.

## Figures and Tables

**Figure 1 foods-12-02366-f001:**
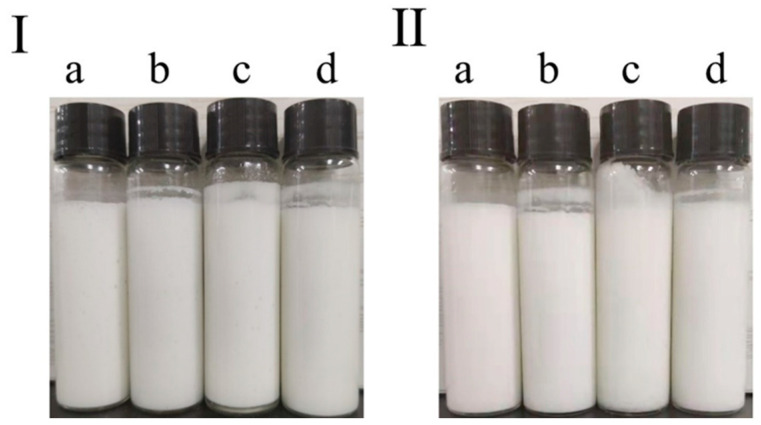
The storage stability of Pickering emulsions formed with β-CD (**I**) and CA/β-CD composite (**II**) with corn oil (**a**), camellia oil (**b**), lard (**c**), and herring oil (**d**) after a storage period of 28 days at room temperature.

**Figure 2 foods-12-02366-f002:**
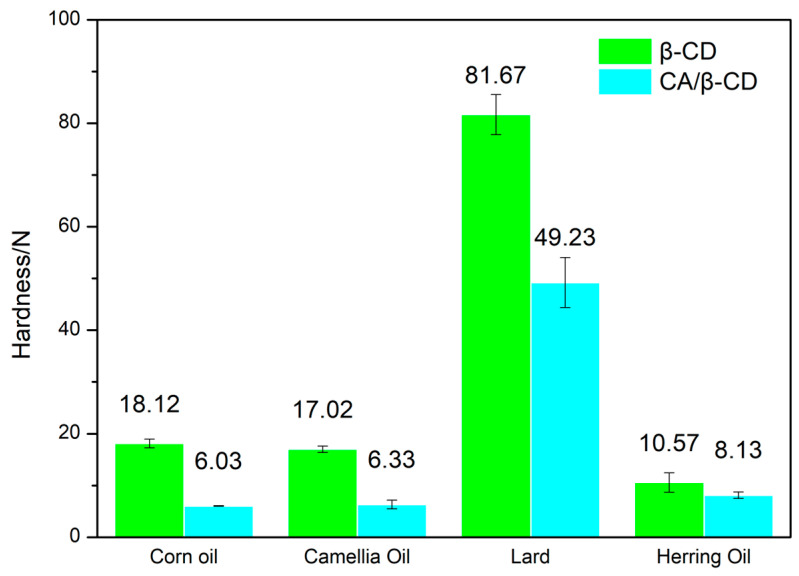
The hardness difference between different Pickering emulsions.

**Figure 3 foods-12-02366-f003:**
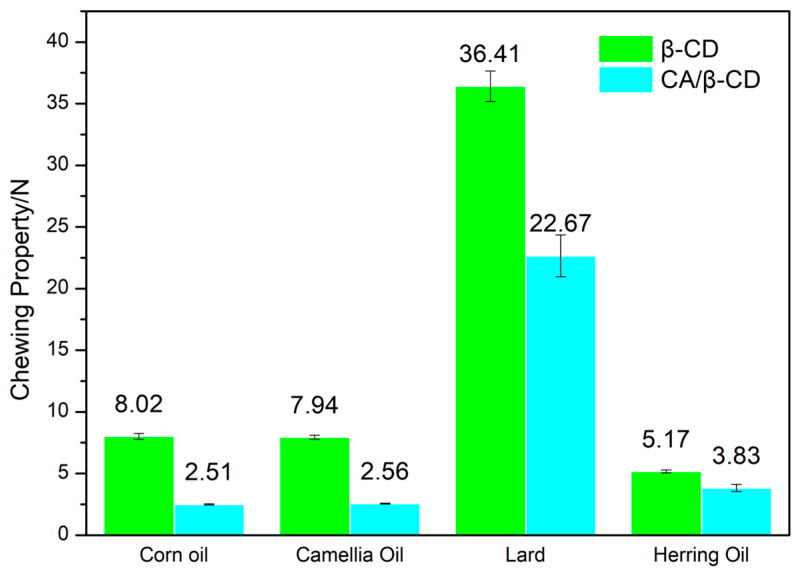
The chewing-property difference between different Pickering emulsions.

**Figure 4 foods-12-02366-f004:**
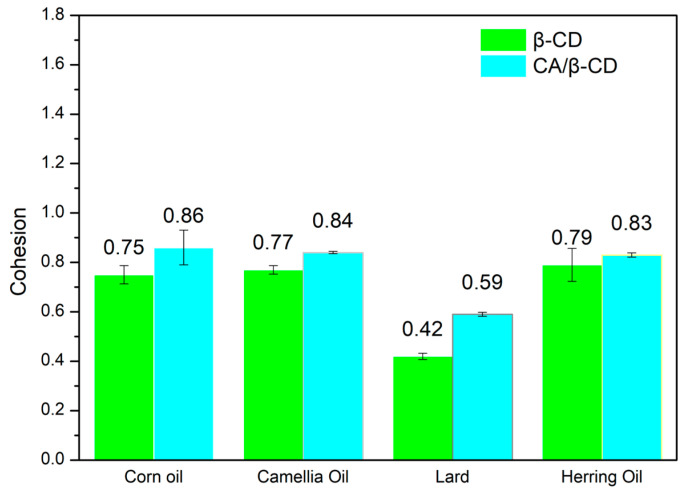
The cohesion difference between different Pickering emulsions.

**Figure 5 foods-12-02366-f005:**
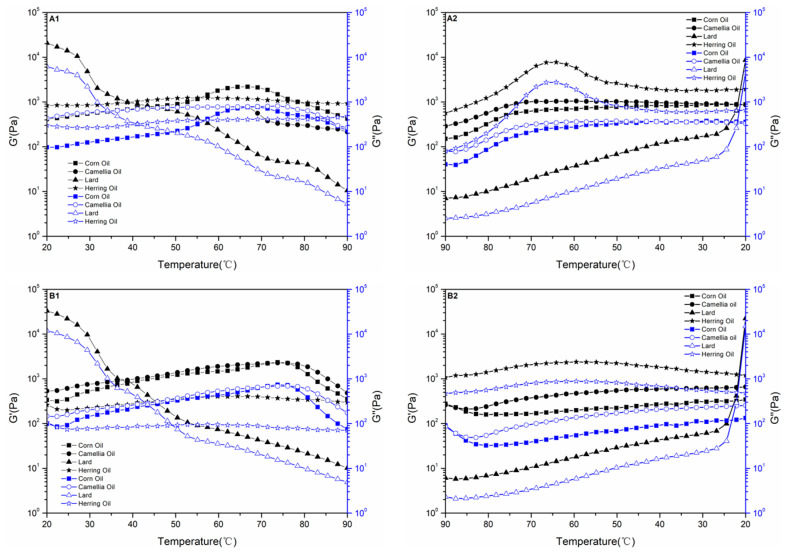
Effect of temperature change on the rheological properties of Pickering emulsion. (**A1**) G′ and G″ diagrams of the β-CD-stabilized Pickering emulsion when the temperature increases. (**A2**) G′ and G″ diagrams of the β-CD-stabilized Pickering emulsion when the temperature decreases. (**B1**) CA/β-CD G′ and G″ of the compound-stabilized Pickering emulsion when the temperature is rising. (**B2**) G′ and G″ of the CA/β-CD-compound-stabilized Pickering emulsion when the temperature is falling.

**Figure 6 foods-12-02366-f006:**
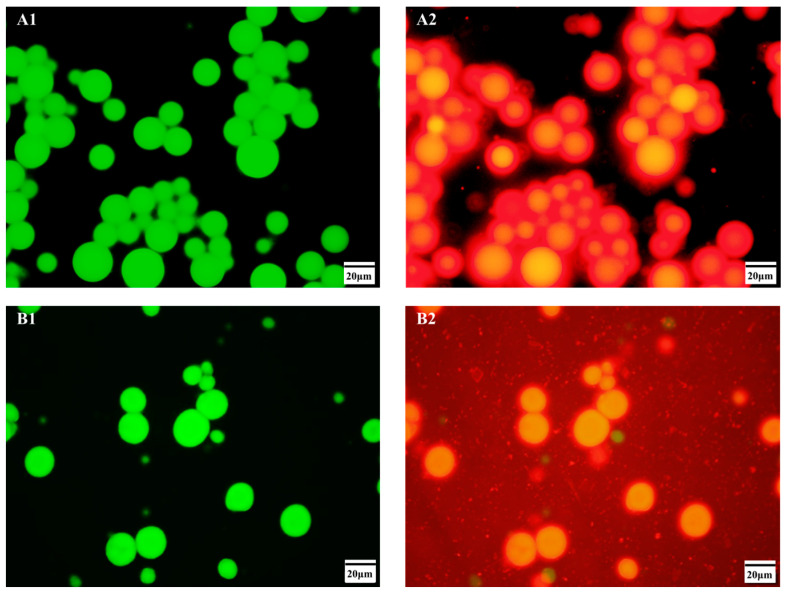
Fluorescence staining micrographs of the emulsions prepared by β-CD (**A**) and CA/β-CD composites (**B**). (**A1**,**B1**) are oil droplets, (**A2**,**B2**) are β-CD and CA/β-CD composite particles, respectively. The oil phase is corn oil.

**Figure 7 foods-12-02366-f007:**
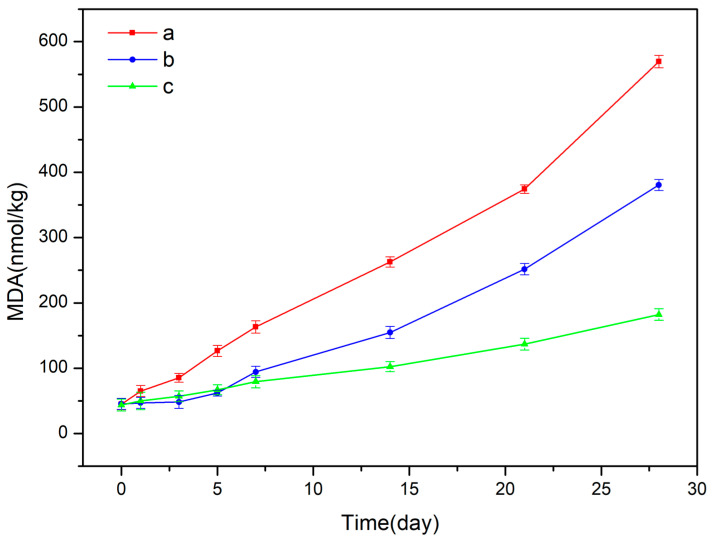
MDA content of Pickering emulsions after 28 days storage at 45 °C: (**a**) β-CD, (**b**) β-CD + CA, and (**c**) CA/β-CD composite. Oil phase is corn oil.

**Figure 8 foods-12-02366-f008:**
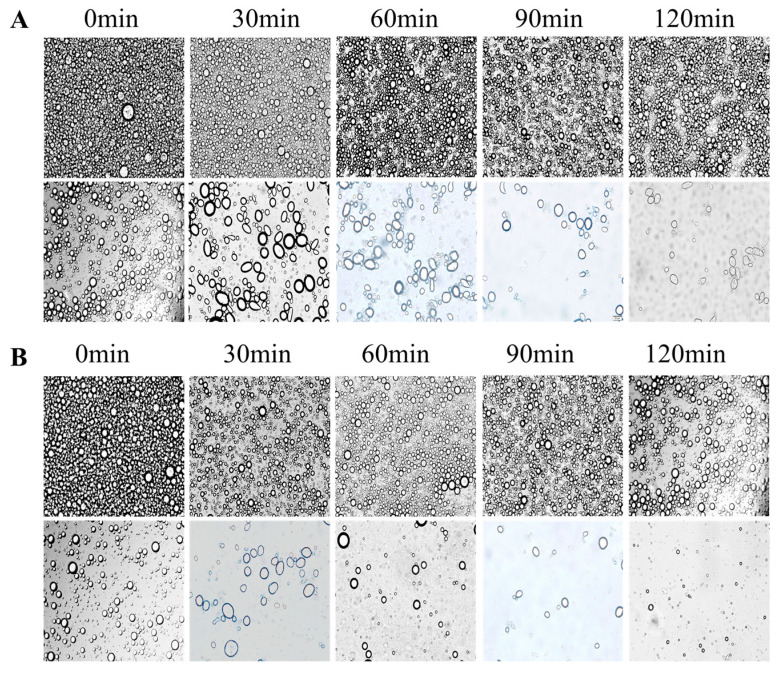
Microstructure observation of Pickering emulsions stabilized in β-CD (**A**) and CA/β-CD (**B**) composites in SGF (top row) and SIF digestion (bottom row) fluid at 0, 30, 60, 90, and 120 min.

**Figure 9 foods-12-02366-f009:**
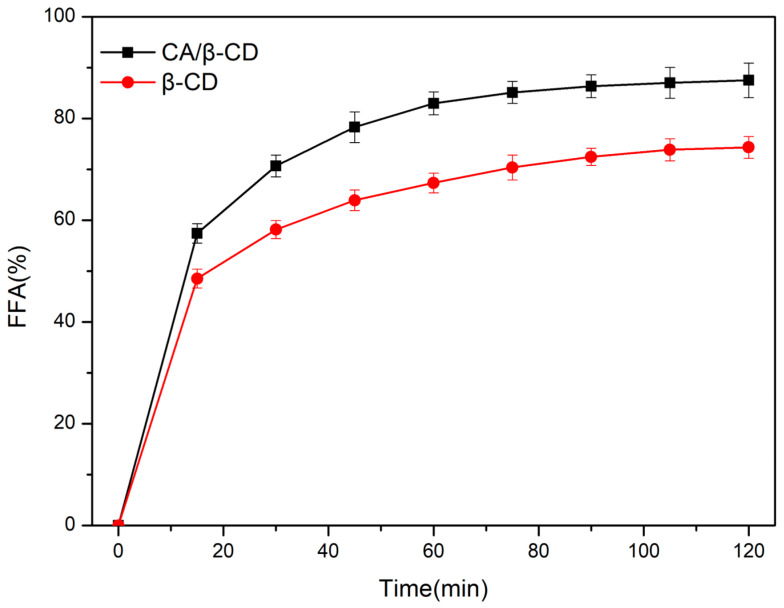
FFA release rate of β-CD- and CA/β-CD-composite-stabilized Pickering emulsion in SIF digestion fluid.

**Table 1 foods-12-02366-t001:** Droplet size (effective diameter) and ζ-potential of the emulsions stabilized by β-CD and the CA/β-CD composite (*n* = 3, mean ± SD).

Sample	Effective Diameter (μm)	ζ-Potential (mV)
β-CD (Corn Oil)	10.52 ± 0.75 ^b^	−32.31 ± 2.47 ^b^
β-CD (Camellia Oil)	11.92 ± 0.63 ^b^	−30.56 ± 0.89 ^b^
β-CD (Lard)	25.57 ± 5.87 ^a^	−23.20 ± 1.32 ^a^
β-CD (Herring Oil)	10.86 ± 1.15 ^b^	−31.33 ± 1.66 ^b^
CA/β-CD (Corn Oil)	11.02 ± 0.45 ^b^	−32.36 ± 1.57 ^b^
CA/β-CD (Camellia Oil)	11.67 ± 0.40 ^b^	−33.40 ± 1.04 ^b^
CA/β-CD (Lard)	27.39 ± 6.67 ^a^	−21.37 ± 1.71 ^a^
CA/β-CD (Herring Oil)	10.67 ± 0.87 ^b^	−29.67 ± 1.37 ^b^

Note: Values in each column with different superscript letters (a and b) are significantly different (*p* < 0.05).

## Data Availability

The data presented in this study are available in “Pickering Emulsion Stabilized by β-Cyclodextrin and Cinnamaldehyde/β-Cyclodextrin Composite”.

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
