# Peer review of "Pickering Emulsion Stabilized by β-Cyclodextrin and Cinnamaldehyde/β-Cyclodextrin Composite"

_foods, 2023, doi:10.3390/foods12122366_

Round 1

Reviewer 1 Report

Thai manuscript is on the preparation and Characterization of Pickering Emulsions using corn oil, camellia oil, lard and herring oil and β-Cyclodextrin and cinnamaldehyde and their complexes as emulsifiers.

The title of manuscript should be revised to show the exact detail of study instead of being a review paper title and covering all aspect of the Pickering Emulsions. In the abstract oil names (corn oil, camellia oil, lard and herring oil) should be added. Don't use "we" word in the writing, and the manuscript style should be more scientific and to do that, need English editing. Abstract should be informative presenting some data, for example for superior thermal stability and palatability.

In introduction, there should be some introduction in the beginning and introduce the emulsions instead of writing directly the disadvantage of traditional emulsions.

It could be better to test the peroxide value along with TBA. Please clarify it. AS in the beginning of the oxidation, the PV rises and then MAD formation gets noticeable.

Writing like "the results were shown in Figure 2." OR "As shown" they are not scientific writing.

Chewing property and Cohesion determination should be mentioned in the method section.

As it is defined in Figure 3 and 4, this should be done in Figure 6 and 7 that what kind of oil was in the formulations.

In discussion part, there is little comparing of the obtained data with previously published data.

Moderate editing of English language

Author Response

Comment 1: Thai manuscript is on the preparation and Characterization of Pickering Emulsions using corn oil, camellia oil, lard and herring oil and β-Cyclodextrin and cinnamaldehyde and their complexes as emulsifiers.

Answer: Thank you very much for your comments.

Comment 2: The title of manuscript should be revised to show the exact detail of study instead of being a review paper title and covering all aspect of the Pickering Emulsions. In the abstract oil names (corn oil, camellia oil, lard and herring oil) should be added. Don't use "we" word in the writing, and the manuscript style should be more scientific and to do that, need English editing. Abstract should be informative presenting some data, for example for superior thermal stability and palatability.

Answer: Your constructive comments are greatly appreciated. In the revised manuscript, we have revised the title, the word "we" has been removed and the sentence has been edited. At the same time, relevant data are supplemented in the Abstract. Please see lines 2-3, 9-10, 12-21 etc.

Comment 3: In introduction, there should be some introduction in the beginning and introduce the emulsions instead of writing directly the disadvantage of traditional emulsions.

Answer: Your constructive comments are greatly appreciated. We supplement the relevant information in the Introduction section. Please see lines 30-38 in the revised manuscript.

Comment 4: It could be better to test the peroxide value along with TBA. Please clarify it. AS in the beginning of the oxidation, the PV rises and then MAD formation gets noticeable.

Answer: Thanks for your advice. MDA as a secondary oxidation product is sufficient to reflect the oxidation resistance of different emulsions. Regarding your suggestions, we will conduct in-depth research in the next research.

Comment 5: Writing like "the results were shown in Figure 2." OR "As shown" they are not scientific writing.

Answer: Thanks for your advice. Based on your suggestion we have edited this in the revised manuscript. Please see lines 274-277,332-334.

Comment 6: Chewing property and Cohesion determination should be mentioned in the method section.

Answer: Thanks for your advice. According to your suggestion, we have supplemented the relevant content. Please see lines 141-145 in the revised manuscript.

Comment 7: As it is defined in Figure 3 and 4, this should be done in Figure 6 and 7 that what kind of oil was in the formulations.

  Answer: Thank you for your suggestion. The method section mentions what kind of oil to use. Please see lines 156-157, 164-165, 186-187, in the revised manuscript.

Comment 8: In discussion part, there is little comparing of the obtained data with previously published data.

Answer: Your constructive comments are greatly appreciated. In the revised manuscript, a number of relevant references have been added. Please see lines 222-223, 232-237, 301-302, 331-334 etc.

Reviewer 2 Report

Article: “Preparation and Characterization of Food-grade Pickering Emulsion with Thermodynamic Stability and Oxidation Resistance”

-          -  It is not clear or not discussed what is the new and difference between this work and the previous work published under the title “Pickering emulsions stabilized by β-cyclodextrin and cinnamaldehyde essential oil/β-cyclodextrin composite” (reference no.1). In this work the author used the term "antioxidant activity" instead “stabilization”.

-        -   “Cinnamaldehyde (CA) is a hydrophobic active matter extracted from natural cinnamon with volatility and special fragrance”. It is isolated and not extracted. Cinnamon is the common name for the used part of the plant. Add the scientific name of the plant species as natural source of CA.

-          - “The antioxidant activity of the emulsion was evaluated by the content of secondary oxide malondialdehyde (MDA) in the emulsion”. Has the emulsion an antioxidant activity without cinnamaldehyde.

it can be improved

Author Response

 Comment 1: It is not clear or not discussed what is the new and difference between this work and the previous work published under the title “Pickering emulsions stabilized by β-cyclodextrin and cinnamaldehyde essential oil/β-cyclodextrin composite” (reference no.1). In this work the author used the term "antioxidant activity" instead “stabilization”.

Answer: Your constructive comments are greatly appreciated. We explain how this paper differs from our previous work in the revised manuscript. Please see lines 94-97.

-     Comment 2: “Cinnamaldehyde (CA) is a hydrophobic active matter extracted from natural cinnamon with volatility and special fragrance”. It is isolated and not extracted. Cinnamon is the common name for the used part of the plant. Add the scientific name of the plant species as natural source of CA.

Answer: Your constructive comments are greatly appreciated. Based on your suggestion, we have edited the relevant content in the revised manuscript. Please see lines 79-82.

-      Comment 3: “The antioxidant activity of the emulsion was evaluated by the content of secondary oxide malondialdehyde (MDA) in the emulsion”. Has the emulsion an antioxidant activity without cinnamaldehyde.

Answer: Thank you very much for your advice. According to references, β-CD cannot improve Pickering emulsion's antioxidant activity. We have corrected the manuscript. Please see lines 419-421.

Reviewer 3 Report

The present paper investigates the properties of 60% (v/v) oil-in-water Pickering emulsions. Two different types of particles have been studied for stabilization of the prepared emulsions – β-cyclodextrin (β-CD) and β-CD/cinnamaldehyde complexes (CA/β-CD) at 3% (w/v) level. The preparation of the CA/β-CD complexes has been studied previously by these authors and is not object of study in the present paper. The main novel findings in the present study are that CA/β-CD complexes can be used for preparation of emulsions with different oily phases (4 different oils have been tested) and that these emulsions have higher oxidation resistance compared to the emulsions prepared with β-CD alone.

Major comments/questions:

-        The title of the paper claims that the prepared emulsions have “thermodynamic stability”. Such statement is not included in the conclusions section. What does the authors had in mind when including this phrase in the title? The thermodynamic stability refers to formation of systems, which are in their lowest energetic state. However, it is well known that most of the emulsions (except for microemulsions) are kinetically stable, not thermodynamically.

-        The present study does not include any information about the drop sizes in the prepared systems. However, as the authors have written in several places within the manuscript – the main oxidation processes are known to proceed at the oil-water interface. Therefore, by changing the total area of this interface, one would also change the possibility for oxidation. Respectively, the decreased content of malondialdehyde shown by the authors for emulsions with CA/β-CD particles may be not due to the inclusion of CA, but due to the formation of bigger in size droplets in these systems and respectively – presence of smaller interfacial area. This hypothesis needs to be carefully considered and reliable data able to exclude it need to be provided in the paper.

-        Experiments with different oils: in their previous study (Ref. 1), the authors have already shown that CA/β-CD complexes can be used for Pickering stabilization of oil-in-water emulsions. In the present study – 4 other oils are used in Sections 3.1-3.3, but in the next sections it is not disclosed which one of the oils have been used to obtain the presented results. Furthermore, except for the lard, which contains higher percentage of saturated fatty acid residues and it is known to have higher melting point, the properties of the other oils are not described in the paper and the reason why these materials have been used is unknown. The authors should include a summary table showing the main fatty acids in each oil, its melting temperature, etc. and this information should be used to comment about the obtained results.

-        The formation of Pickering emulsions stabilized by β-CD or β-CD-complexes have been extensively studied. The present information included in the introduction section needs to be edited to include all relevant references, see for example Kibichi, Kahveci, J. Food. Sci 2019 (10.1111/1750-3841.14619); a recent review paper by Jug et al., 10.1007/s10847-021-01097-z and other papers in the field.

-        There are many claims within the paper, which need to be précised and supported with relevant references and/or results – see for example lines 36-39; 182; claim in lines 192-194 – it has been already shown for numerous systems in the literature; 242-244; 258-288; 300-301; 305-311

-        The discussion about possible crystallization of the drops, emulsifiers, its dissolution, desorption of the particles, etc… in sections 3.2 and 3.3 needs to be edited and the conclusions made by the authors need to be clearly stated. All different physico-chemical processes which are claimed to “may happen” – should be studied and the final explanation for the observed results should be given.

-        Procedures described in sections 2.3.6 and 2.3.7 – are these originally developed by the authors for this study or are replicated as previously published elsewhere? Appropriate references should be included. The coefficients used in the formulas, lines 148 and 169, need to be explained.

Minor specific comments/questions:

-        Corn oil – it is claimed to be purchased from two different sources – see lines 84 and 90 – what is the reason for this? What is the difference between the two sources?

-        The type of homogenizer used for emulsions preparation is not written in the paper.

-        The rheological measurements are performed at 1 mm gap. What is the reason for this? Has the wall slip between the plates and the sample been considered in these measurements?

-        The procedures for determination of the chewing properties and cohesion are not currently explained.

-        Figure 5 needs to be improved. What is the reproducibility of these measurements – for example Fig5A2 – star symbols? What is the reason for the differences observed between liquid oils?

-        The present definition for creaming index gives a value of 100% creaming, although the emulsions remain stable. Maybe – the authors should reconsider the definition and instead of the present one, use the frequently used one with lower serum volume/emulsion height, which would give 0% creaming for their samples.

Overall, the paper is relatively clearly written.

However, there are some repetitions which need to be edited - see for example lines 129-131; 137-139; 152-154. Also, there are several places in which the text is written in non-sentence format: see for example lines 12-13; 67-69; 103-106, etc.

Also, lines 191-192; 255, 256; 260-261 needs to be edited.

Author Response

Major comments/questions:

Comment 1: The title of the paper claims that the prepared emulsions have “thermodynamic stability”. Such statement is not included in the conclusions section. What does the authors had in mind when including this phrase in the title? The thermodynamic stability refers to formation of systems, which are in their lowest energetic state. However, it is well known that most of the emulsions (except for microemulsions) are kinetically stable, not thermodynamically.

Answer: Your constructive comments are greatly appreciated. The original title wants to expresses that within a certain range of changes (20-65°C), Pickering emulsion is stable. As you mentioned, the title of the paper is not appropriate. We greatly appreciate your suggestion and have decided to revise the thesis title. Please read the title of the paper in the revised manuscript.

Comment 2: The present study does not include any information about the drop sizes in the prepared systems. However, as the authors have written in several places within the manuscript – the main oxidation processes are known to proceed at the oil-water interface. Therefore, by changing the total area of this interface, one would also change the possibility for oxidation. Respectively, the decreased content of malondialdehyde shown by the authors for emulsions with CA/β-CD particles may be not due to the inclusion of CA, but due to the formation of bigger in size droplets in these systems and respectively – presence of smaller interfacial area. This hypothesis needs to be carefully considered and reliable data able to exclude it need to be provided in the paper.

Answer: Thank you for your suggestion. In the revised manuscript we have provided the droplet size of the emulsion, see lines 136-140, 239-369. Pickering emulsions prepared by β-CD and CA/β-CD had similar sizes under the same oil phase conditions. Thus, the main reason for the difference in oxidation resistance of emulsions is the introduction of CA at the oil-water interface, rather than a change in droplet size.

Comment 3: Experiments with different oils: in their previous study (Ref. 1), the authors have already shown that CA/β-CD complexes can be used for Pickering stabilization of oil-in-water emulsions. In the present study – 4 other oils are used in Sections 3.1-3.3, but in the next sections it is not disclosed which one of the oils have been used to obtain the presented results. Furthermore, except for the lard, which contains higher percentage of saturated fatty acid residues and it is known to have higher melting point, the properties of the other oils are not described in the paper and the reason why these materials have been used is unknown. The authors should include a summary table showing the main fatty acids in each oil, its melting temperature, etc. and this information should be used to comment about the obtained results.

Answer: Thank you for your suggestion. The method section mentions what kind of oil to use. Please see lines 152-153, 160-161, 182-183, in the manuscript. The properties of other oils we have added in the revised manuscript. Please see lines 255-260.

Comment 4: The formation of Pickering emulsions stabilized by β-CD or β-CD-complexes have been extensively studied. The present information included in the introduction section needs to be edited to include all relevant references, see for example Kibichi, Kahveci, J. Food. Sci 2019 (10.1111/1750-3841.14619); a recent review paper by Jug et al., 10.1007/s10847-021-01097-z and other papers in the field.

Answer: Thank you for your suggestion. We have supplemented the article content and references in the manuscript. Please see lines 30-38, 73-75, 79-82 etc.

Comment 5: There are many claims within the paper, which need to be précised and supported with relevant references and/or results – see for example lines 36-39; 182; claim in lines 192-194 – it has been already shown for numerous systems in the literature; 242-244; 258-288; 300-301; 305-311

Answer: Thank you for your suggestion. Your suggestions have made our article better and we have added references based on your suggestions. Please see lines 46-48, 223-224, 235-238, 313-315, 330-359, 370-371, 375-379 etc.

Comment 6: The discussion about possible crystallization of the drops, emulsifiers, its dissolution, desorption of the particles, etc… in sections 3.2 and 3.3 needs to be edited and the conclusions made by the authors need to be clearly stated. All different physico-chemical processes which are claimed to “may happen” – should be studied and the final explanation for the observed results should be given.

Answer: Thank you for your suggestion. We have provided a more detailed explanation of the experimental phenomenon in the revised manuscript based on your suggestion. Please see sections 3.2 and 3.3.

Comment 7: Procedures described in sections 2.3.6 and 2.3.7 – are these originally developed by the authors for this study or are replicated as previously published elsewhere? Appropriate references should be included. The coefficients used in the formulas, lines 148 and 169, need to be explained.

Answer: Thanks for your suggestion. In sections 2.3.6 and 2.3.7 - they are replicated as previously published elsewhere. Relevant references have been included in the manuscript. Please see lines 171-172, 190-191.

For the explanation of the coefficients in the formulas in lines 148 and 169, please see lines 183-185, 207-210.

Minor specific comments/questions:

Comment 8: Corn oil – it is claimed to be purchased from two different sources – see lines 84 and 90 – what is the reason for this? What is the difference between the two sources?

Answer: Thank you very much for your careful reading, corn oil was obtained from Kele Fine Chemical Co., Ltd. (Hubei, China). The occurrence of corn oil in line 90 was due to our clerical error and we have corrected it in the manuscript.

Comment 9: The type of homogenizer used for emulsions preparation is not written in the paper.

Answer: Thank you very much for your advice. We have supplemented the type of homogenizer in the article, please see lines 126-127.

Comment 10: The rheological measurements are performed at 1 mm gap. What is the reason for this? Has the wall slip between the plates and the sample been considered in these measurements?

Answer: The samples we prepared all have gel properties, and no wall slip occurs between the plates and the sample. Rheological measurements at 1 mm gap can save samples.

Comment 11: The procedures for determination of the chewing properties and cohesion are not currently explained.

Answer: Thanks for your advice. According to your suggestion, we have supplemented the relevant content, please see lines 142-146.

Comment 12: Figure 5 needs to be improved. What is the reproducibility of these measurements – for example Fig5A2 – star symbols? What is the reason for the differences observed between liquid oils?

Answer: The data can be repeated as many times as necessary. The difference between liquid oils may be due to the ratio of saturated and unsaturated fatty acids.

Comment 13: The present definition for creaming index gives a value of 100% creaming, although the emulsions remain stable. Maybe – the authors should reconsider the definition and instead of the present one, use the frequently used one with lower serum volume/emulsion height, which would give 0% creaming for their samples.

Answer: Your constructive comments are greatly appreciated. The original manuscript has been edited. Please see lines 134 and 229.

Comment 14: Comments on the Quality of English Language

Overall, the paper is relatively clearly written.

However, there are some repetitions which need to be edited - see for example lines 129-131; 137-139; 152-154. Also, there are several places in which the text is written in non-sentence format: see for example lines 12-13; 67-69; 103-106, etc.

Answer: Thank you very much for your suggestion, we have edited the sentences in the manuscript.

Comment 15: Also, lines 191-192; 255, 256; 260-261 needs to be edited.

Answer: Thank you very much for your suggestion, we have edited the sentences in the manuscript.

Reviewer 4 Report

The manuscript is interesting and well written. In my opinion, the references chosen to justify some aspects are not as well chosen as they could be. In addition, some more techniques could be added that would make the characterization even better, but I think that what is provided could be enough if average drop diameters can be estimated.

I consider that a more general reference than that given should be used to define Pickering emulsions. Is it a necessary condition not to have surfactant to be Pickering? Or is the necessary condition that the predominant stabilizing mechanism is the presence of solid particles?

There are other very recent studies of pickering emulsions with fumed silica (Aerosil 200 for example) or zeina, a preference is detected to include references of the same nationality when there are other manuscripts equal or more related to this and in the same journals. 

Indicate the equipment used for sample preparation.

Why haven't flow curves been carried out?

Are the authors sure that if G´ is greater than G ́ ́ the samples have gel-like character?

Being the droplet size distributions, droplet mean sizes and polydispersity, some of the most important variables in the characterization of emulsions, why have they not been determined? Can not be estimated sizes using microscopy since laser diffraction has not been used?

Author Response

Comments and Suggestions for Authors

The manuscript is interesting and well written. In my opinion, the references chosen to justify some aspects are not as well chosen as they could be. In addition, some more techniques could be added that would make the characterization even better, but I think that what is provided could be enough if average drop diameters can be estimated.

A: We appreciate your recognition of our work. In the revised manuscript we have evaluated the droplet size. Please see lines 135-139 and 238-268.

I consider that a more general reference than that given should be used to define Pickering emulsions. Is it a necessary condition not to have surfactant to be Pickering? Or is the necessary condition that the predominant stabilizing mechanism is the presence of solid particles?

A: Thanks for your advice. In Pickering emulsions, solid particles are the predominant stabilizing mechanism. In the manuscript we have made corrections. Please see lines 39-40.

There are other very recent studies of pickering emulsions with fumed silica (Aerosil 200 for example) or zeina, a preference is detected to include references of the same nationality when there are other manuscripts equal or more related to this and in the same journals.

A: Thanks for your advice. The reference here has been revised based on your suggestion. Please see lines 42-44 and 49-53.

Indicate the equipment used for sample preparation.

A: Thanks for your advice. In the revised manuscript, we have included the equipment used to prepare the samples. Please see lines 126-127.

Why haven't flow curves been carried out?

A: Our previous article studied the flow curves of Pickering emulsions stabilized by β-CD and CA/β-CD composites. DOI: https://doi.org/10.1016/j.foodchem.2021.131995. This study focuses on the effect of temperature change on emulsions.

Are the authors sure that if G´ is greater than G ́ ́ the samples have gel-like character?

A: I’m sure that if G´ is greater than G ́ ́ the samples have gel-like character. We have added this in the revised manuscript. Please see lines

Being the droplet size distributions, droplet mean sizes and polydispersity, some of the most important variables in the characterization of emulsions, why have they not been determined? Can not be estimated sizes using microscopy since laser diffraction has not been used?

A: Your constructive comments are greatly appreciated. In the revised manuscript we have evaluated the droplet size. Please see lines 135-139 and 238-268.

Round 2

Reviewer 1 Report

Comments were included in the revised version of the manuscript, and its acceptance is suggested.
